# Dual activities of an X-family DNA polymerase regulate CRISPR-induced insertional mutagenesis across species

Trevor Weiss [1,2,3,4,7], Jitesh Kumar [1,2,3,4,7], Chuan Chen [5], Shengsong Guo [1,2], Oliver Schlegel[6], John Lutterman[6], Kun Ling [5] & Feng Zhang [1,2,3,4] ✉

The canonical non-homologous end joining (c-NHEJ) repair pathway, generally viewed as stochastic, has recently been shown to produce predictable outcomes in CRISPR-Cas9 mutagenesis. This predictability, mainly in 1-bp insertions and small deletions, has led to the development of in-silico prediction programs for various animal species. However, the predictability of CRISPR-induced mutation profiles across species remained elusive. Comparing CRISPR-Cas9 repair outcomes between human and plant species reveals significant differences in 1-bp insertion profiles. The high predictability observed in human cells links to the template-dependent activity of human Polλ. Yet plant Polλ exhibits dual activities, generating 1-bp insertions through both templated and non-templated manners. Polλ knockout in plants leads to deletion-only mutations, while its overexpression enhances 1-bp insertion rates. Two conserved motifs are identified to modulate plant Polλ's dual activities. These findings unveil the mechanism behind species-specific CRISPR-Cas9-induced insertion profiles and offer strategies for predictable, precise genome editing through c-NHEJ.

Clustered regularly interspaced short palindromic repeats (CRISPR)-Cas-mediated genome editing, a potent tool for introducing intended genetic modifications in both fundamental and translational research, operates through the sequential steps of searching, binding, and cleaving the targeted sequences, guided by programmable CRISPR RNA[1–3]. Subsequent repair of double-stranded DNA cleaved breaks can be accomplished via either end-joining or homology-directed pathways. Generally, the end joining pathways, including canonical or alternative non-homologous end joining (c-NHEJ or a-NHEJ), are considered more efficient but imprecise, resulting in stochastic insertion or deletion (indel) events. In contrast, the homology-directed repair (HDR) pathway is commonly employed to achieve precise editing outcomes, such as base substitutions or precise insertions/deletions, albeit with much less efficiency[4,5], Despite substantial efforts to enhance its efficiency, achieving precise genome editing through the HDR pathway remains challenging in many species[6].

An alternative strategy to achieve predictable and precise genome editing has been explored by harnessing the end-joining pathways. Although generally considered imprecise, the end-joining pathways have been utilized to facilitate the knock-in of DNA templates or deletion of intervening sequences between two cleavage sites[7–10]. Furthermore, recent studies indicated that the indel outcomes induced by the CRISPR-*Sp*Cas9 (i.e. *Streptococcus pyogenes* Cas9; hereafter referred to as Cas9) system exhibited predictable

[1]Department of Plant and Microbial Biology, University of Minnesota, Saint Paul, MN 55108, USA. [2]Center for Precision Plant Genomics, University of Minnesota, Saint Paul, MN 55108, USA. [3]Microbial and Plant Genomics Institute, University of Minnesota, Minneapolis, MN 55108, USA. [4]Center for Genome Engineering, University of Minnesota, Minneapolis, MN 55108, USA. [5]Department of Biochemistry and Molecular Biology, Mayo Clinic, Rochester, MN 55905, USA. [6]Department of Genetics, Cell Biology and Development, University of Minnesota, Minneapolis, MN 55455, USA. [7]These authors contributed equally: Trevor Weiss, Jitesh Kumar. ✉e-mail: zhangumn@umn.edu

patterns[8,11,12]. By targeting a large set of CRISPR-Cas9 sites in human and mouse cells, it was observed that up to 47% of the sites produced a single indel type[12]. These findings led to the development of in-silico programs, such as FORECasT and InDelphi, with remarkable accuracy in predicting the end joining-mediated indel mutations in various species, including yeast, zebrafish, Xenopus, mouse, and human[11–14].

Among the predominant indel mutations induced by CRISPR-Cas9, 1-bp insertions were identified as the most commonly occurring events[11,12]. The templated insertion model has been proposed to explain the 1-bp insertions observed in yeast and animal species[13]. In this model, Cas9 generates a 1-nt 5′ overhang by cleaving the −3rd position on the CRISPR RNA complementary strand and the −4th position on the non-complementary strand upstream of the Proto-spacer Adjacent Motif (PAM) sequence. Subsequently, the overhangs can be filled in by an X-family DNA polymerase, DNA Pol 4 or λ, and ligated through the c-NHEJ pathway resulting in the duplication of the nucleotide at the −4th position[13,15].

Recent efforts to extend CRISPR-Cas9 genome editing into plants have confirmed the prevalence of 1-bp insertions in various plant species[9,16–18]. While these findings suggested a potential extension of predictability observed in animals to plants, recent studies have also indicated that 1-bp insertion profiles may not be consistently predictable as proposed by the templated insertion model[9,16,17]. The existence of distinctive 1-bp insertion profiles and the underlying mechanism remains unclear. Addressing these questions is crucial for refining predictability and enhancing the end-joining mediated precise genome editing across species.

In this study, we systematically compare the CRISPR-Cas9-induced 1-bp insertion profiles among two plant species, *Arabidopsis thaliana*, representing a dicot model plant, and *Setaria viridis*, representing a monocot model plant, and human cells. Our observations reveal the presence of distinctive 1-bp insertion profiles induced by CRISPR-Cas9 in plants. To delve deeper into the underlying mechanism, we identify the X-family DNA Polymerase, Polλ, responsible for plant-specific 1-bp insertion profiles and demonstrate that the efficiency and profiles of 1-bp insertions can be modulated by dual activities of Polλ. Finally, we propose an updated model to account for distinctive CRISPR-induced 1-bp insertions. Overall, our findings provide a thorough evaluation of the predictability of CRISPR-Cas9 mutagenesis and shed light on the intricate interplay between CRISPR-Cas9 and DNA repair pathways across species. This research underscores that the prediction tools for CRISPR mutagenesis need to be tailored and optimized for each individual species, accounting for the inherent differences in the DNA repair machinery. Our work will also pave the way for the development of more precise and predictable genome editing strategies using the c-NHEJ pathway.

## Results

### Low predictive power of CRISPR-Cas9 mutagenesis prediction programs for plants

To assess the predictability of the CRISPR-Cas9-induced mutations in plants, we examined the performance of two widely used CRISPR mutagenesis prediction programs, FORECasT and InDephi[11,12]. We generated CRISPR-induced mutations at 59 sites, including 26 from *Arabidopsis* and 33 from *Setaria*, by introducing the corresponding CRISPR-Cas9 constructs into each species (Supplementary Data 1). In *Arabidopsis*, each CRISPR-Cas9 construct was introduced using the floral dip-based stable transgenic approach. Individual seedlings of each T1 transgenic plant were collected for the CRISPR mutation analysis at each target site. In *Setaria*, Individual CRISPR-Cas9 constructs were transformed via transient protoplasts transfection. Transformed protoplast cells were collected after 48 h for the mutation assay. Subsequently, the mutation profile of each site was obtained by using the next-generation sequencing (NGS) based assay. The indel mutagenesis rates averaged at 8.9% and 28.4% at the

sites from *Arabidopsis* and *Setaria*, respectively (Supplementary Fig. 1a). Additionally, the indel profile from each site was further characterized into individual insertion and deletion types for each species (Fig. 1a). Notably, the 1-bp insertions represent one of the most common occurring mutation types, as previously observed in human cell lines. In *Arabidopsis*, 1-bp insertions were the most prevalent mutation types, accounting for an average of 44.6% of all mutations across 26 CRISPR sites (Fig. 1a). For *Setaria viridis*, the average 1-bp insertion rate appeared to be the 4th highest at 9.6% across 33 CRISPR sites (Fig. 1a).

Simultaneously, the predicted mutation profile was generated for each target site using FORECasT and InDephi. In this study, we chose to focus on the insertion rates for correlation analyses on the predicted versus observed values for the following reasons: (1) CRISPR-induced insertions appeared to exhibit less stochastic patterns than deletions; and (2) previous studies have suggested that these prediction tools demonstrate greater predictive power for insertions compared to other indel types[11,12,14]. As a result, we observed no positive correlations using either FORECasT or InDelphi for both *Arabidopsis* and *Setaria* datasets (Fig. 1b, c). Weak negative correlations were observed in the *Arabidopsis* dataset ($r = -0.56$, $p < 0.0031$ and $r = -0.4$, $p < 0.036$; Fig. 1b), while no correlation was found in the *Setaria* dataset ($r = -0.18$, $p < 0.31$ and $r = -0.07$, $p < 0.69$; Fig. 1c). Thus, our data suggested that both prediction programs developed with human datasets exhibited low predictive power for the CRISPR-Cas9-induced mutation profile in plants.

### Distinctive templated versus non-templated 1-bp insertion patterns across species

The limited predictive power from the human cell-based indel prediction tools prompted us to further examine CRISPR-Cas9-induced insertion profiles in plants. Both FORECasT and InDelphi predicted CRISPR-induced insertions primarily as 1-bp insertion events occurred at the −4th position upstream of the PAM; and most of these insertions were derived from templated insertions by duplicating the −4th nucleotide. When we analyzed the observed insertions from the *Arabidopsis* and *Setaria* target sites, 1-bp insertions were consistently predominant, accounting for averagely 95.9% of insertions across all sites (Supplementary Fig. 1b). However, when the 1-bp insertion patterns were plotted according to the −4th nucleotide, the observed insertions did not consistently exhibit characteristics of templated insertions in plants (Fig. 2a). When the −4th nucleotide was T, the inserted nucleotide appeared to follow the templated insertion model with 78.8% and 75.7% of insertions as T in *Arabidopsis* and *Setaria*, respectively (Fig. 2a). With the −4th nucleotide as A, while A remained the predominant inserted nucleotide (58.5% and 58.7% in *Arabidopsis* and *Setaria*), the fractions of other types of insertions, termed as non-templated insertions, increased substantially (Fig. 2a). In cases where the −4th nucleotide was either C or G, non-templated insertions became predominant by increasing to 61.4% and 66.0% for the −4th nucleotide C, and 98.4% and 99.5% for the −4th nucleotide G in *Arabidopsis* and *Setaria*, respectively.

To compare with 1-bp insertion patterns in human cells, we analyzed the insertion profiles of 150 target sites previously reported from the human cell lines[13]. The results were largely consistent with the templated insertion model, showing the 1-bp insertion pattern with the −4th nucleotide duplications, while low levels of non-templated 1-bp insertions were observed at the target sites with −4th nucleotide as C or G (Fig. 2a). Taken together, our observations revealed distinct 1-bp insertion patterns between plants and human cells. The 1-bp insertion profiles from plant species exhibited a higher incidence of non-templated insertions, deviating from the templated insertion model. Notably, the rates of non-templated insertions appeared to vary depending on the −4th nucleotide, increasing in the order of T, A, C, and G.

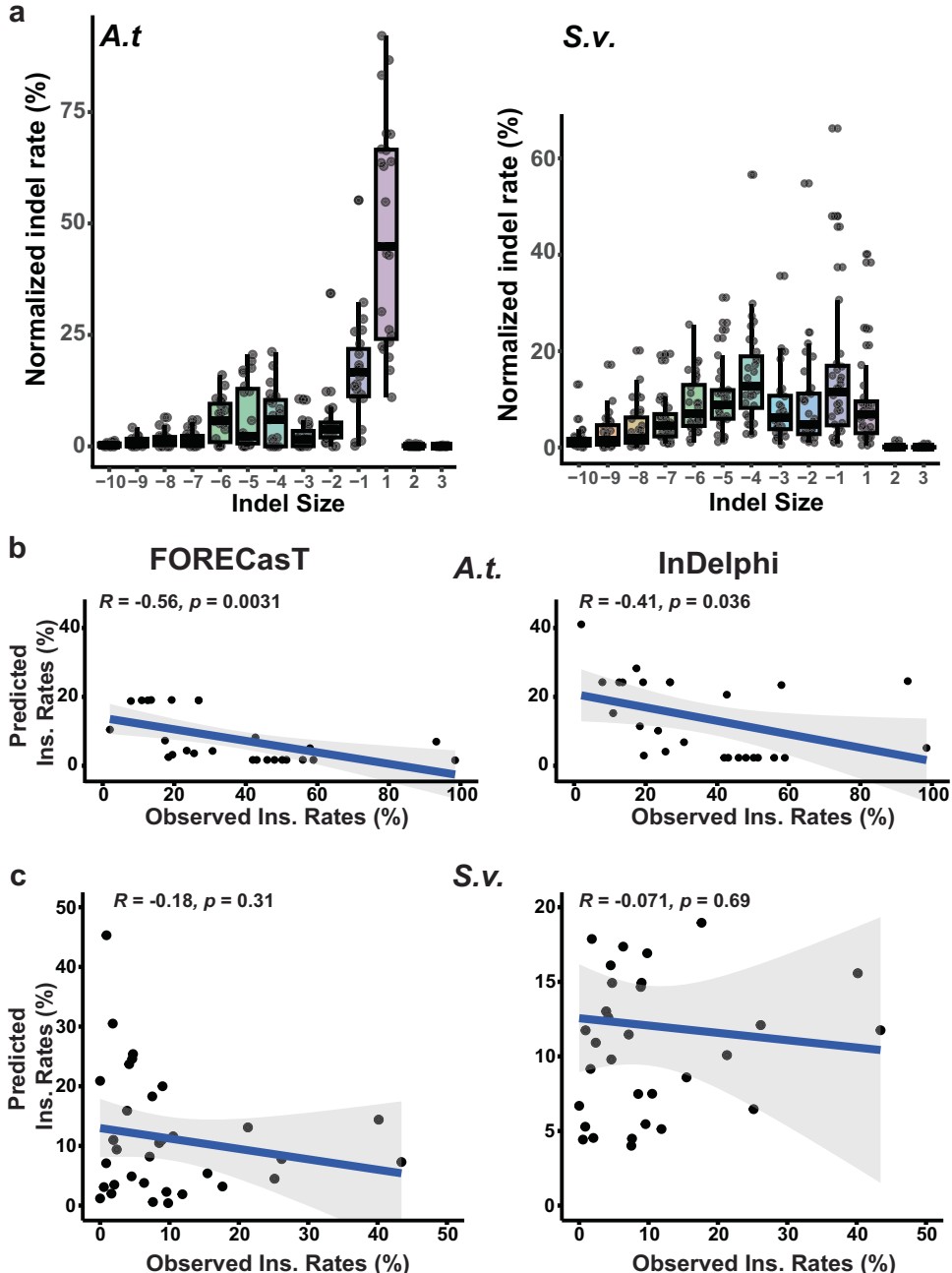

**Fig. 1 | Predictability of FORECasT and InDelphi tools for CRISPR-Cas9 induced insertions in plants. a** CRISPR-Cas9 induced mutation profiles across 59 target sites in *Arabidopsis* (*n* = 26) and *Setaria* (*n* = 33). *X*-axis represents individual indel sizes. The normalized mutation rates (*Y*-axis) were determined by dividing the number of reads containing mutations within each indel size by the total number of reads containing all types of mutations. The horizontal bars within boxes represent medians. The top and bottom edges of the boxes represent the 75th and 25th

percentiles, respectively. The upper and lower whiskers extend to data no more than 1.5x the interquartile range from the upper and lower edges of the box, respectively. **b**, **c** Two-sided Pearson correlation analysis were performed using scatter plots to compare predicted versus experimentally observed insertion (ins.) rates for each CRISPR gRNA in *Arabidopsis* (*n* = 26) and *Setaria* (*n* = 33). The 95% confidence interval (CI) were indicated with gray color. The source data are provided in the Source Data file.

## Plant-specific 1-bp insertion profiles dependent on the −4th nucleotide context

To further explore the distinctive 1-bp insertion profiles across species, we conducted direct comparisons by targeting identical CRISPR sites in *Arabidopsis*, *Setaria*, and human cell line, HEK293. This involved initially integrating the firefly luciferase gene and subsequently expressing the CRISPR-Cas9 expression cassette in the genomes of these three species (Fig. 2b). We designed two CRISPR guide RNAs (gRNAs) to target overlapping sites located on opposite strands, referred to as inverted PAM (iPAM) targets, as described in previous research[13] (Fig. 2c). These

two gRNAs, with one −4th position as T (iPAM_T) and the other as G (iPAM_G), represented the sequence contexts for the highest and lowest templated insertion rates observed in plants (Fig. 2c). In *Arabidopsis*, CRISPR-Cas9 constructs were assembled with the firefly luciferase reporter gene in T-DNA. The resulting constructs were transformed using the floral dip-based stable transgenic approach. Three seedlings from each T1 transgenic group were collected for CRISPR mutation analysis at each target site. For *Setaria viridis*, a homozygous *Setaria* line with the firefly luciferase reporter gene integrated into the genome was obtained from previous research[19]. Individual CRISPR-Cas9 constructs

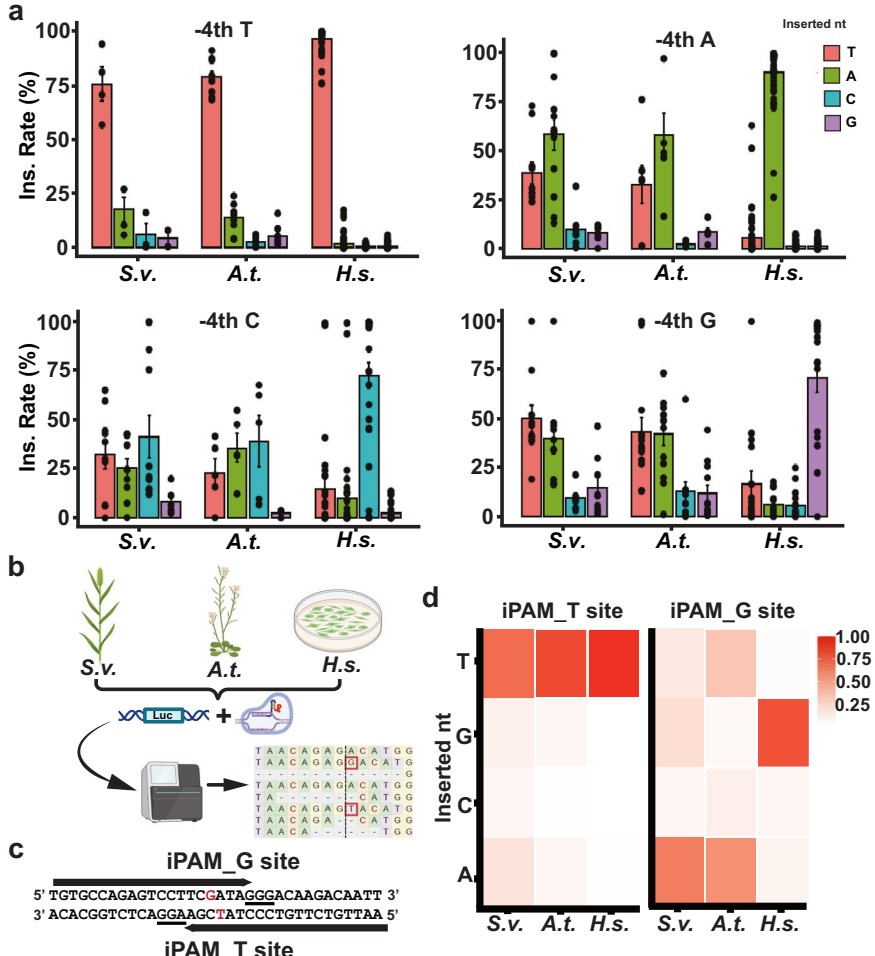

**Fig. 2 | Cross-species analyses of templated versus non-templated 1-bp insertions according to the −4th nucleotide. a** Cross-species 1-bp insertion patterns to the −4th nucleotide. The 1-bp insertions were divided into 4 groups according to the inserted nucleotide for each CRISPR gRNA. The normalized 1-bp insertion (ins.) rates were calculated by dividing the number of reads containing each type of 1-bp insertions by the number of reads with all types of 1-bp insertions and were plotted to the −4th nucleotides (T, A, C, and G) for *Setaria viridis* (*S.v.*; $n = 33$ biologically independent samples), *Arabidopsis thaliana* (*A.t.*; $n = 26$ biologically independent samples), and the human cell line (*H.s.*; $n = 150$ biologically independent samples).

Data are presented as mean values ± SEM. **b**. The schematic workflow to compare 1-bp insertion patterns across *S.v.*, *A.t.*, and *H.s.* line through the next-generation sequencing assay. **c** The CRISPR targeted sequences of iPAM_T and G. The −4th nucleotide was highlighted in red with the PAM sequence underlined. **d** Heatmap analyses of the proportion of each inserted nucleotide type (nt) at the −4th position of the iPAM_T and G sites across *S.v.* ($n = 3$), *A.t.* ($n = 3$), and *H.s.* ($n = 3$). The source data are provided in the Source Data file. **b** Created with BioRender.com released under a Creative Commons Attribution-NonCommercial-NoDerivs 4.0 International license.

were then transformed into protoplast cells isolated from the luciferase gene-containing plants. Transformed protoplasts were collected after 48 h for the mutation assay with 3 replications for each target site. When insertion rates were examined, both CRISPR gRNAs induced substantial 1-bp insertions ranging from 33.8% to 89.0% for the iPAM_T site and from 33.4% to 74.4% for the iPAM_G site in three species (Supplementary Fig. 2).

Next, we analyzed templated versus non-templated insertion patterns at each target site. In the HEK293 cells, consistent with the templated insertion model, templated insertions were predominantly presented at both target sites with rates of 97.0% and 84.8%, respectively (Fig. 2d). However, in *Arabidopsis* and *Setaria*, predominant templated insertions were primarily observed at the iPAM_T site, ranging from 73.3% to 87%. At the iPAM_G site, non-templated insertions were predominant, accounting for 72.6% to 95.8% of 1-bp insertions in both plant species (Fig. 2d). Taken together, these findings corroborated the observations from 59 individual target sites, revealing distinct plant-specific 1-bp insertion profiles. These profiles exhibited either templated or non-templated dominant patterns associated by the −4th nucleotide upstream of PAM.

## Limited influence of chromatin states on 1-bp insertion profiles in plants

As indicated by prior studies, both epigenetic and genetic factors could influence CRISPR-Cas9 induced mutation profiles[15,17,20]. To explore the mechanism underlying the distinctive 1-bp insertion profiles in plants, we investigated the impact of the chromatin states on these insertions. We used the multi-copy CRISPR target site (MCSite) system previously developed in *Arabidopsis*[17]. Two sets of MCSites, designated as MCSite_T and MCSite_G based on their −4th nucleotide, are located in diverse epigenetic contexts as described previously[17]. Individual sites within each MCsite family can be categorized into two major groups as either open and unmethylated or closed and methylated chromatin (Fig. 3a, b).

When the 1-bp insertion rates of individual MCSites were examined, variations were found across different chromatin states as previously indicated[17]. For MCSite_T sites, insertion rates ranged from 7.9% to 26.8%, and for MCSite_G sites, they ranged from 41.9% to 58.9% (Fig. 3a, b). In contrast, heatmap analysis of the 1-bp insertion profiles revealed a consistent pattern within each MCSite family. Specifically, for MCSite_T sites, templated insertions were predominantly observed

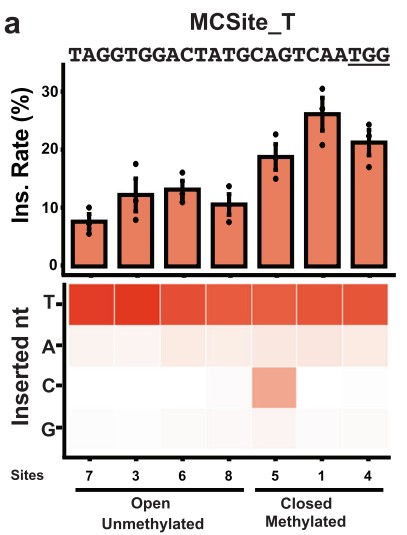
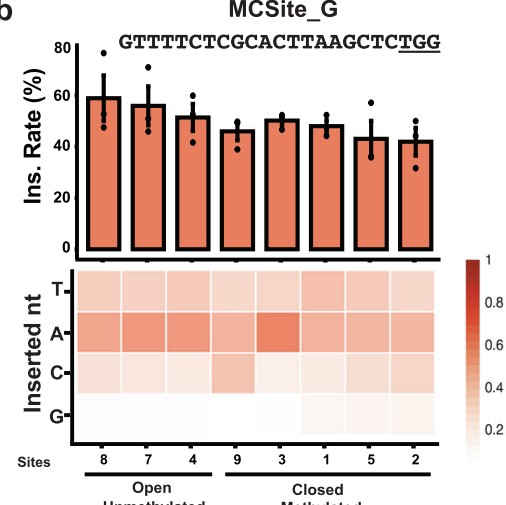

**Fig. 3 | Effect of chromatin states on the 1-bp insertion rate and profile in *Arabidopsis*.** Normalized 1-bp insertion rates were plotted for individual MCsite_T (**a**) and MCsite_G (**b**) sites (X-axis). The 20-bp targeted sequences with 3-bp underlined PAM sequences were shown with each plot. The normalized 1-bp insertion (ins.) rates were determined by dividing the number of reads containing 1-bp insertions by the total number of reads containing all types of indel mutations. Data are presented as mean values ± SEM from three independent plants. Heatmaps in the lower panel illustrated the proportion of each inserted nucleotide type (T, A, C, G) at the −4th position of individual MCsite_T (**a**) and MCsite_G (**b**) sites. Chromatin states of individual sites were categorized into Open and Unmethylated or Closed and Methylated groups. The source data are provided in the Source Data file.

across individual sites, regardless of their chromatin states (Fig. 3a). On the other hand, all individual sites within the MCSite_G family exhibited a predominant 1-bp non-templated insertion pattern across different epigenetic contexts, ranging from 94.0% to 99.8% (Fig. 3b). These results suggested that chromatin states may have limited impacts on CRISPR-Cas9 induced 1-bp insertion profile.

## Polλ homolog responsible for both templated and non-templated 1-bp insertions in plants

We then investigated the genetic factors contributing to the distinct 1-bp insertion profiles in plants. Previous studies have pointed to the X-family DNA polymerase, Polλ, and its homolog as pivotal players in mediating 1-bp templated insertions in human and yeast cells[13,15]. A single copy of the Polλ homolog was identified in both *Arabidopsis* and *Setaria* genomes through sequence homology searches[21]. No other X-family DNA polymerases were found in plants from the homology search. Phylogenetic analyses confirmed that this plant X-family DNA polymerase exhibited a close evolutionary relationship with Polλ as opposed to other members, such as DNA Pol μ and Terminal deoxynucleotidyl Transferase (TdT) (Supplementary Fig. 3 and Supplementary Data 2).

To explore the involvement of the plant Polλ homolog in CRISPR-Cas9 induced 1-bp insertions, we obtained an *Arabidopsis* T-DNA knock-out mutant line (*atpolλ-1*), previously characterized with no notable growth or physiological defects[22,23]. Using the wild type and the homozygous *atpolλ-1* mutant *Arabidopsis* plants, we generated stable transgenic plants with the CRISPR-Cas9 T-DNA construct to target three distinct sites: the single-copy site in the *Arabidopsis Chelatase I2* gene (*AtCHLI2*), as well as the MCSite_T and MCSite_G sites. Three T1 CRISPR-Cas9 transgenic plants from each genotype were used to survey CRISPR-induced mutations for each target site. The single-copy CHLI2 site would allow for a rapid assessment of the involvement of Polλ in 1-bp insertions, while the two MCSites provided additional insights in different epigenetic contexts.

When we examined CRISPR-Cas9 mutagenesis at the CHLI2 site, both wild-type and mutant CRISPR-Cas9 plants displayed comparable overall mutagenesis rates, averaging 38.9% and 37.9%, respectively (Fig. 4a). In wild-type plants, approximately 25.3% of indel mutations were identified as 1-bp insertions at the −4th position, with non-templated insertions being predominant at a rate of 65.2%, attributable

to the G nucleotide at the −4th position in the CHLI2 site (Fig. 4b; Supplementary Fig. 4a). In contrast, in Polλ mutant plants, the 1-bp insertion rates, encompassing both non-templated and templated insertions, were reduced to undetectable levels (0.2%; Fig. 4b; Supplementary Fig. 4b). Additionally, we explored the potential involvement of this Polλ homolog in CRISPR-Cas9 induced deletions. As a result, we observed similar levels of deletions within three different deletion groups, 1-bp, 2 to 10-bp and more than 10-bp, between the wild-type and mutant plants (Fig. 4c). Thus, the plant Polλ homolog appeared to be the pivotal gene for CRISPR-Cas9 induced 1-bp insertions, operating in both templated and non-templated manners, with limited involvement in deletions.

Furthermore, we investigated the role of this Polλ homolog at additional CRISPR target sites within diverse epigenetic contexts. When examining the 1-bp insertion rates at the MCSite_T and G sites, we observed significant reductions of 1-bp insertions, both templated and non-templated, across all sites, irrespective of their chromatin states. In the MCSite_T sites, the 1-bp insertion rates decreased from an average of 19.5% in wild-type plants to 1.6% in the mutant plants, while in the MCSite_G sites, the rates were reduced from an average of 49.4% to 1.8% (Fig. 4d, e). These results substantiated that the plant Polλ homolog is responsible for both templated and non-templated 1-bp insertions regardless of chromatin states.

## Overexpression of AtPolλ restores or enhances templated and non-templated 1-bp insertions

Next, we hypothesized that overexpression of Atpolλ could restore or even enhance the 1-bp insertion rates. To test this hypothesis, we generated stable transgenic plants by overexpressing the AtPolλ gene in the *atpolλ-1* mutant plants. The AtPolλ coding sequence was driven under the constitutive *Arabidopsis Ubiquitin-10* promoter and cloned into the final construct with a CRISPR-Cas9 expression cassette to target the CHLI2 and MCSite_T sites. Three T1 CRISPR-Cas9 transgenic plants with the *atpolλ-1* mutant genotype were used to survey CRISPR-induced mutations for each target site. When 1-bp insertions were examined at the CHLI2 site, the AtPolλ overexpression plants exhibited a 1.6-fold increase compared to wild-type plants, with an average rate of 39.8% (Fig. 5a). The 1-bp insertion profiles appeared similar between the AtPolλ overexpression and the wild-type plants, with non-

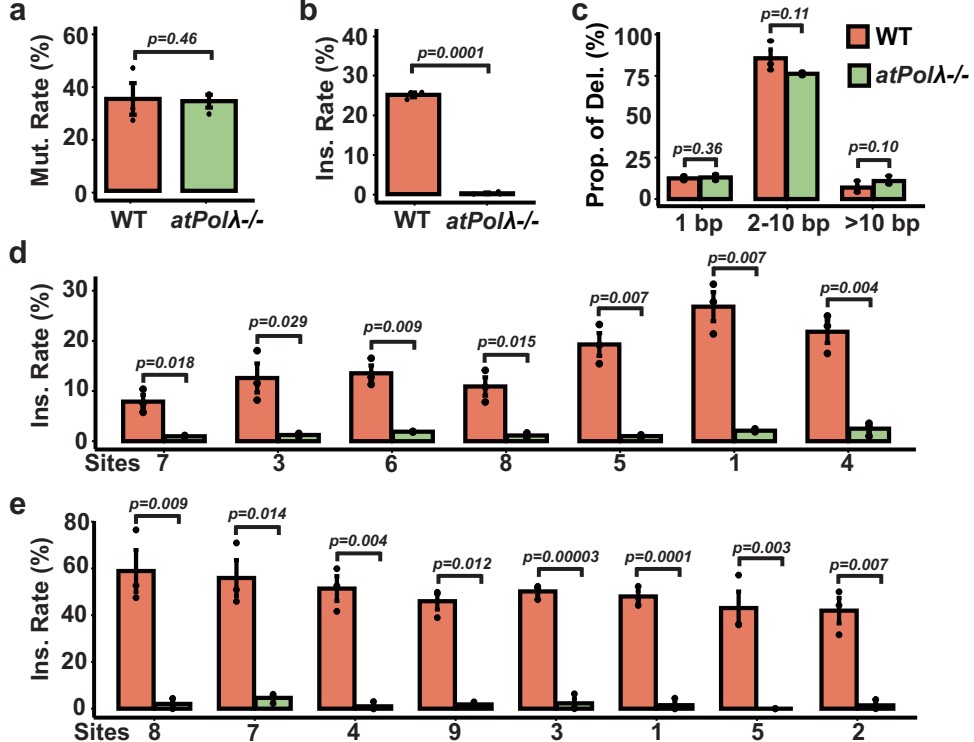

**Fig. 4 | Polλ is responsible for 1-bp insertions in *Arabidopsis*. a** CRISPR-Cas9 mutation (mut.) rates between the wild type and *atpolλ*-1 mutant plants. The mutation rates (*Y*-axis) were determined by dividing the number of reads containing indel mutations by the total number of NGS reads. **b** Normalized 1-bp insertion rates between the wild type and *atpolλ*-1 mutant plants at the CHLI2 site. The normalized 1-bp insertion (ins.) rates were determined by dividing the number of reads containing 1-bp insertions by the total number of reads containing all types of indel mutations. **c**. Normalized deletion rates between the wild type (WT) and *atpolλ*-1 mutant plants at the CHLI2 site. The normalized proportion of deletion (Prop. of Del. as *Y*-axis) were determined by dividing the number of reads containing deletions within each category (1-bp, 2-10 bp, or >10 bp) by the total number of reads containing all types of deletions. **d**, **e** Normalized 1-bp insertion (ins.) rates between the wild type and *atpolλ*-1 mutant plants at the MCsite_T (**d**) and MCsite_G (**e**) sites. Heatmaps under the bar plots illustrate the proportion of each inserted nucleotide type (T, A, C, G) at the −4th position of individual MCsite_T (**d**) and MCsite_G (**e**) sites. Data are presented as mean values ± SEM from 3 independent plants. *P*-values were derived from unpaired one-tailed Student's *t* test. The source data are provided in the Source Data file.

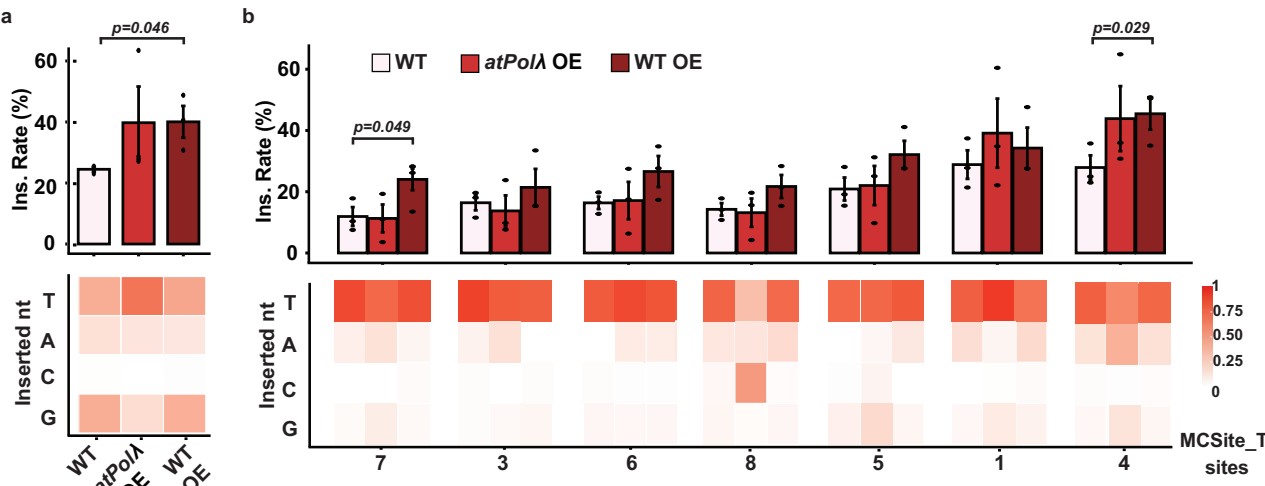

**Fig. 5 | Overexpression of Polλ restores or enhances 1-bp insertions in *Arabidopsis*.** Normalized 1-bp insertion rates at CHLI2 (**a**) and MCsite_T (**b**) among three lines: wild-type plants (WT), Polλ overexpression plants in the *atpolλ*-1 mutant (*atpolλ* OE), and Polλ overexpression plants in the wild-type backgrounds (WT OE). The normalized 1-bp insertion rates (*Y*-axis) were determined by dividing the number of reads containing 1-bp insertions by the total number of reads containing all types of indel mutations. Heatmaps under each plot illustrated the proportion of each inserted nucleotide type (T, A, C, G) at the −4th position. Data are presented as mean values ± SEM from 3 independent plants. *P*-values were derived from unpaired one-tailed Student's *t* test. The source data are provided in the Source Data file.

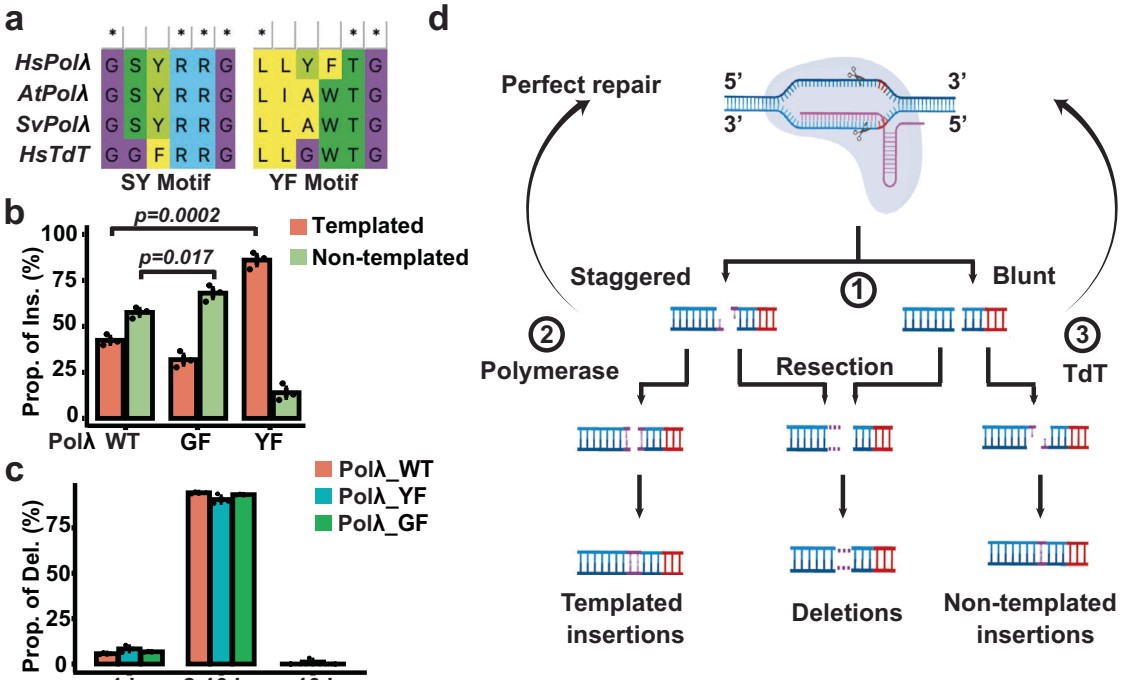

**Fig. 6 | Dual activities of AtPolλ are modulated by two conserved motifs.**
**a** Sequence alignment of two conserved motifs, SY and YF, across Human Polλ, AtPolλ, SvPolλ, and human TdT. **b** Comparisons of templated versus non-templated insertion rates between the wild type AtPolλ and two variants, Polλ$^{S366G/Y367F}$ and Polλ$^{A459Y/W460F}$ at the CHLI2 site. The templated (indicated by orange) or non-templated insertion (indicated by green) rates (Y-axis) were determined by dividing the number of reads containing each type of 1-bp insertions by the total number of reads containing 1-bp insertions in each sample. **c** Normalized deletion rates between the wild type and two variants at the CHLI2 site. The normalized deletion rates (Y-axis) were determined by dividing the number of reads containing deletions within each category (1-bp, 2-10 bp, or >10 bp) by the total number of reads containing all types of deletions. Data are presented as mean values ± SEM from three independent plants. *P*-values were derived from unpaired one-tailed Student's *t* test. The source data are provided in the Source Data file. **d** The proposed

model for the dual activities of Polλ in generating templated and non-templated 1-bp insertions. **Step 1**: CRISPR-Cas9 generates a blunt or staggered cut at the targeted site. Blunt-ended cleavages occur at the -3rd position upstream of the PAM (indicated by the red vertical lines) on both strands, while staggered cleavages take place with one cut at the −4th position on the non-targeted strand and the other cut at the -3rd position on the targeted strand, producing 5' 1-nt overhangs. **Step 2**: The staggered product can be filled in by Polλ with template-dependent activity. **Step 3**: The blunt-ended product can be processed by Polλ with template-independent activity to extend 1-nt at the 3' end of each strand. After ligation and correction by c-NHEJ and mismatch repair, non-templated 1-bp insertions occur at the −4th position. Additionally, cleavage products could be processed through either perfect ligation, indicated by the curved arrowheads, or through resection to generate deletion, indicated by the purple dash lines.

templated insertions still being predominant at an average rate of 74.8% (Fig. 5a). When examining the 1-bp insertions at the MCSite_T sites, overexpression of the AtPolλ transgene in the mutant plant appeared to restore 1-bp insertion rates to the levels observed in wild-type plants at five of seven MCSite_T sites. At the other 2 sites, sites 1 and 4, the 1-bp insertion rates exhibited substantial increases by 1.4 to 1.6 folds, respectively (Fig. 5b). When comparing the 1-bp insertion profiles, similar insertion patterns were observed between the overexpression and wild-type plants with predominant templated insertions across nearly all the sites except for one site, site 8 (Fig. 5b). These results confirmed that overexpression of AtPolλ could restore or may enhance CRISPR-Cas9 induced templated and non-templates 1-bp insertions in the knockout mutant plants, further validating its pivotal role in generating 1-bp insertions.

We further hypothesized that overexpression of this gene should have the potential to enhance 1-bp insertions in wild-type plants. To test this idea, we introduced the same overexpression construct to wild-type plants. Three T1 CRISPR-Cas9 transgenic plants with the wild-type background were used to survey CRISPR-induced mutations for each target site. At the CHLI2 site, we observed a similar increase in the 1-bp insertion rate between the overexpression wild-type plants and the overexpression mutant plants compared to the wild-type control plants (Fig. 5a). At the MCSite_T sites, when comparing the 1-bp insertion rates between the overexpression wild-type plants and the wild-type control plants, we observed substantial increases in all seven

sites by 1.2 to 2.0 folds (Fig. 5b). When comparing the 1-bp insertion profiles, similar insertion patterns were observed between the overexpression wild-type plants, the overexpression mutant plants, and the wild-type control across all the sites, irrespective of their epigenetic states (Fig. 5b). Taken together, these observations corroborated that overexpressing the Polλ homolog in wild-type plants could further increase 1-bp insertions.

**Conserved motifs modulate the dual activities of AtPolλ**
To gain insights into the mechanism(s) underpinning the distinct properties of Polλ across species, we conducted protein sequence analyses by aligning AtPolλ with X-family DNA Polymerases in humans (Supplementary Fig. 5). Previous studies have indicated two conserved motifs in human X-family DNA Polymerase that contribute to template dependency[24]. The first motif, identified as GSYRRG in template-dependent human DNA polymerases λ, features two amino acids, serine and tyrosine (SY), which are replaced by glycine and phenylalanine (GF) in the template-independent human TdT (Fig. 6a and Supplementary Fig. 5)[24,25]. The second motif, known as the YF motif, contains tyrosine and phenylalanine at the catalytically active sites of the DNA polymerases λ. In contrast, these two residues are changed to glycine and tryptophan (GW) in TdT (Supplementary Fig. 5)[24,25]. When analyzing these motifs in DNA polymerase λ homologs from *Arabidopsis*, *Setaria*, Tobacco, and rice, the first motif was identical to the sequences in human Polλ, while the second motif, characterized by

alanine and tryptophan (AW), showed a closer resemblance to the GW motif found in human TdT (Fig. 6a and Supplementary Fig. 5). Thus, the plant Polλ homologs appear to combine characteristic motifs from human Polλ and TdT.

The presence of both human Polλ and TdT motifs could potentially contribute to the observed dual templated-dependent and independent activities in AtPolλ. We then hypothesized that the dual activities of AtPolλ could be modulated by modifying each motif individually. To test this hypothesis, we generated two variants of AtPolλ through site-directed mutagenesis on the respective motifs. The first variant, AtPolλ^YF, was engineered by substituting Alanine and Tryptophan (AW) with Tyrosine and Phenylalanine (YF) at the second motif to mimic human Polλ (Fig. 6a). Similarly, the second variant, AtPolλ^GF, was created to mimic human TdT by replacing Serine and Tyrosine (SY) with Glycine and Phenylalanine (YF) at the first motif (Fig. 6a).

The coding sequence of each AtPolλ variant was cloned into the T-DNA vector described above, with the constitutive *Arabidopsis Ubiquitin-10* promoter and a CRISPR-Cas9 expression cassette to target the CHLI2 site. We used an *agrobacterium*-mediated transient expression approach to transform individual T-DNA constructs into young seedlings of the *atpolλ* knock-out mutant, and then examined the CRISPR-Cas9 mutation profile at the CHLI2 site using the NGS assay (Supplementary Fig. 6a). The average mutation rates from these samples are 17.3% (AtPolλ^WT), 12.7% (AtPolλ^GF) and 10.3% (AtPolλ^YF), respectively (Supplementary 6b). When analyzing templated versus non-templated 1-bp insertion patterns, the samples expressing the wild type AtPolλ gene exhibited higher proportions of non-templated insertions compared to those of templated insertions (57.6% non-templated insertions versus 42.4% templated insertions) consistent with the observations from the stable transgenic plants (Fig. 5a, b). In contrast, the samples transformed with the AtPolλ^YF variant demonstrated altered 1-bp insertion profiles with templated insertion proportions being significantly higher than those from the overexpression of the wildtype AtPolλ by 100% (86.0% versus 42.4%; Fig. 6b and Supplementary Fig. 6c, d). Conversely, the samples transformed with the AtPolλ^GF variant displayed significantly higher proportions of non-templated insertions compared to the wild-type AtPolλ over-expression lines by 18% (67.9 % versus 57.6%; Fig. 6b and Supplementary Fig. 6c, d). Regarding the deletion profiles, no evident differences were observed within three different deletion groups, 1-bp, 2 to 10-bp and more than 10-bp, among AtPolλ^WT and the two variants (Fig. 6c).

Notably, the overall 1-bp insertion rates from the samples with each variant reduced to 4.4% and 5.5% compared to 31.7% in the wild-type AtPolλ overexpression control, suggesting the involvement of additional amino acids in regulating enzymatic activity. (Supplementary Fig. 6e). Collectively, these observations align with our hypothesis that these two conserved motifs play crucial roles in modulating the dual template-dependent and independent activities of AtPolλ. Further investigation is required to refine the enzymatic activities of these variants.

## Discussion

In this study, we initially investigated CRISPR-Cas9-induced insertion profiles across different species and subsequently delved into the mechanism responsible for plant-specific 1-bp insertion patterns. Our findings revealed that prediction tools developed with human datasets demonstrated limited accuracy in predicting insertional mutations in plants. This discrepancy seemingly stemmed from plant unique 1-bp insertion patterns. Specifically, in human cells, templated 1-bp insertions were predominantly found at most CRISPR-Cas9 target sites, while both templated insertions and non-templated 1-bp insertions could occur at substantial levels in plants, inconsistent with the templated insertion model. Our observations highlighted the significance of the sequence context, especially −4th nucleotide, in determining

the 1-bp insertion profile. The rates of non-templated insertions exhibited substantial variations based on the type of −4th nucleotide, increasing in the order of T, A, C, to G. In some cases, however, variations could be observed even with the same −4th nucleotide, suggesting the involvement of other factors in determining 1-bp insertion patterns (Fig. 2a). Further investigation will be needed to fully address this question.

We identified the crucial gene responsible for both templated and non-templated 1-bp insertions in *Arabidopsis* as the homolog of a specialized X-family DNA polymerase, Polλ. Disrupting this Polλ homolog resulted in a nearly complete reduction of 1-bp insertions while overexpressing the gene led to the restoration or enhancement of 1-bp insertions. Previous research suggested that plant Polλ homologs may play a role in repairing oxidative base damage, UV-induced DNA repair or general double-strand DNA break repair, akin to their human counterpart[24,26,27]. However, the specific function of plant Polλ has remained elusive[22,23,28]. Our study provided genetic evidence indicating the direct involvement of plant Polλ in the end processing during the c-NHEJ repair. However, unlike human and yeast Polλ genes, which predominantly filled in gaps in a template-based manner, the *Arabidopsis* Polλ homolog seemed capable of processing broken ends through both templated and non-templated mechanisms.

Different levels of template dependence have been observed in X-family DNA polymerases. In human and most vertebrates, the X-family DNA polymerases consist of four members, Polλ, β, μ, and Terminal deoxynucleotidyl Transferase (TdT)[29]. Human Polλ shows a high level of template dependence, while TdT appears to be entirely template-independent[30]. This aligns with the observation that Polλ-mediated 1-bp insertions are mostly template-based in human cells. However, in most plants, only one X-family DNA polymerase, Polλ, is present, with no other X-family paralogs in their genome. One possible explanation for increased levels of non-templated 1-bp insertions in plants could be that the plant Polλ may possess dual activities, catalyzing both template-dependent and -independent reactions. Interestingly, an in vitro biochemical study has suggested that the rice Polλ homolog exhibited both template-based polymerase and non-templated-based TdT activities[28,31]. Furthermore, protein sequence analysis revealed that plant Polλ homologs possess characteristic motifs from both human Polλ and TdT, which could elucidate the observed dual activities of AtPolλ. Our in vivo assay with the AtPolλ variants, generated by modifying these motifs, further supports this hypothesis. The AtPolλ variant resembling human Polλ significantly increased the proportion of templated insertions, while the human TdT-like variant significantly increased the proportion of non-templated insertions. These findings suggest that the dual activities of plant Polλ stem from the combination of conserved motifs from both template-dependent and independent polymerases.

Moreover, it is worth noting that the frequencies of a specific inserted nucleotide can vary for non-templated insertions. For example, lower frequencies of cytosine (C) insertions compared to adenine (A), thymine (T), and guanine (G) were observed in the CHLI2 and MCSite_T sites (Fig. 3b and Supplementary Fig. 4). One possible explanation could be that the AtPolλ exhibits preferences for certain types of nucleotides when installing non-templated insertions. Substrate nucleotide preferences have been previously reported in human Terminal deoxynucleotidyl transferase (TdT), which catalyzes non-templated nucleotide additions. In vitro studies have shown that human TdT exhibits a preference for incorporating nucleotides in the order of G>T = A>C[32,33]. It is plausible that AtPolλ also displays nucleotide preferences, potentially favoring G, T, and A over C during non-templated insertions in vivo. To explore this hypothesis and elucidate the AtPolλ activity during non-templated synthesis, further experiments would be required.

Taking these observations together, we propose that dual activities of the X-family DNA polymerase, Polλ, plays a pivotal role in

determining templated versus non-templated 1-bp insertion patterns (Fig. 6d). According to this model, CRISPR-Cas9 can cleave its target sites in two modes: blunt-ended cleavage at the -3rd position upstream of the PAM and staggered cleavages with one cut at the −4th position on the non-targeted strand and the other cut at the -3rd position on the targeted strand[8,9,13,15,34]. Following cleavage, Polλ homologs with dual activities could use either staggered or blunt products as substrates to generate a spectrum of templated and non-templated insertions. In support of this view, previous studies have indicated that template-dependent DNA polymerase favors sticky DNA ends with overhangs as substrates, while the template-independent TdT prefers blunt ends[30]. Moreover, the ratio between staggered and blunt cleavage can vary in different CRISPR sites. Recent studies in human cell lines have suggested that the sequence context, particularly the nucleotide at the −4 position, could play a crucial role in influencing the ratio of blunt and staggered cleavages. For example, target sites with a thymine (T) at the −4 position appeared to exhibit higher staggered cleavage compared to those with other nucleotides[11,35]. In our model, a higher frequency of staggered cleavages would result in more templated insertions, and vice versa. While the nucleotide at the −4 position has been identified as a key contributing factor, it is likely that additional sequence features or other factors could influence the ratio of blunt and staggered cleavages. To systematically explore this question further, experiments employing approaches to capture double-strand break (DSB) end structures in vivo could be employed[36]. Furthermore, noticeable levels of non-templated 1-bp insertions were also observed from CRISPR sites in human cells particularly when the −4th nucleotides were C or G. It has been suggested that other non-template dependent X-family DNA polymerases, such as Pol or TdT, could contribute to this phenomenon[35]. Further research will be required to systematically address these questions.

It is noteworthy that, in this study, we compared 1-bp insertion profiles across different cell states from three different species, including non-dividing Setaria protoplast cells, dividing cells from Arabidopsis seedlings, and human dividing cell lines. It has been known that cell states, especially cell cycle stages, could influence DNA repair pathway choices and expression of DNA repair genes[37]. To address the potential differences in 1-bp insertion profiles between dividing and non-dividing cells, we conducted an experiment using Arabidopsis protoplasts to target three CRISPR sites: CHLI2, MCsite_T, and MCSite_G. When comparing the 1-bp insertion profiles of these sites between the stable transgenic lines and protoplasts, we observed highly similar 1-bp mutation outcomes between the two groups (Supplementary Fig. 6f, Supplementary Data 4). While our data indicate that the cell cycle state may not be a major determinant of the 1-bp insertion patterns observed in this study, it has been reported that environmental factors can regulate the expression level and protein stability of AtPolλ[38,39]. Further investigation would be required to examine the CRISPR mutagenesis profiles under different growth conditions.

To rigorously test this dual activity model, it will be imperative to further dissect the properties of Polλ homologs through both genetical and biochemical approaches. Nevertheless, our findings unveiled a unique role of a specialized DNA polymerase in regulating DNA repair outcomes, holding several implications for enhancing the predictability and precision of CRISP-Cas9 mediated genome editing. The varying rates and profiles of 1-bp insertions observed across species could stem from distinct properties of Polλ activity or its differential involvement in repairing CRISPR-Cas9-induced double-strand breaks. This underscores how inherent variability in DNA repair pathways and the enzymatic properties of key repair proteins, such as Polλ, contribute to the species-specific CRISPR-Cas9 editing profiles. Consequently, enhancing the predictability of CRISPR-Cas9 mutagenesis prediction tools necessitates tailoring and optimizing these tools for individual species. Moreover, manipulating the activities of Polλ and its

homologs could serve to fine-tune the frequency and profiles of CRISPR-Cas9 mutagenesis. For instance, disrupting Polλ may result in deletion-only CRISPR-Cas9 mutagenesis profiles. Conversely, in scenarios where precise or predictable 1-bp insertions are desirable, strategies involving the overexpression or targeted recruitment of Polλ at specific target sites can be explored. In Arabidopsis plants overexpressing Polλ, we have demonstrated substantial increases in the frequency of 1-bp insertions at CRISPR target sites. Further investigation is warranted on whether directly recruiting Polλ to targeted genomic loci could further augment 1-bp insertion frequencies, thereby improving the predictability of mutagenesis outcomes. This strategy could pave the way for more predictable CRISPR-Cas9 mutagenesis across diverse species by circumventing the intrinsic variability in DNA repair pathway machinery among organisms. As such, it could also present a powerful tool for precisely introducing 1-bp insertions at specific sites in gene therapy, especially considering that over 8000 human disease-related pathogenic alleles result from 1-bp deletions[40].

## Methods

### Plant materials and growth conditions
The Arabidopsis thaliana Columbia ecotype (Col-0) and Setaria viridis (ME034 ecotype) were used in these experiments. Arabidopsis plants were grown in a growth chamber with the following conditions: 16-/8-h light/dark cycle, 22 °C, and 55% humidity. Setaria growth conditions for protoplasting were performed as previously described[19].

Homozygous Arabidopsis Polλ (AT1G10520) knock-out line (atPolλ-1) was obtained from Arabidopsis Biological Resource Center (ABRC) with the stock number SALK_075391C[22]. The mutant plants were grown under the same conditions as the wild-type Col-0 plants.

### Vector construction
Plasmids for Arabidopsis and Setaria transformation and CRISPR-Cas9 mutagenesis experiments were created using the method described previously[41]. In brief, the gRNAs, listed in Supplementary Data 1, were first cloned into pMOD_B2301 (for Arabidopsis; Addgene #91067) or pMOD_B2303 (for Setaria; Addgene #91068) to create gRNA expression plasmids. The Setaria protoplast transfection constructs were generated by using the Golden Gate assembly method with pMOD_A1110 (the Cas9 expressing plasmid, Addgene #91031), the gRNA plasmids, pMOD_C3001 (the GFP reporter plasmid; Addgene #91094), and pTRANS100 (the destination plasmid; Addgene #91198). The Arabidopsis T-DNA transformation constructs were generated using the Golden Gate assembly method with pMOD_A0101 (the Cas9 expressing plasmid, Addgene #90998), the gRNA plasmids, pMZ105 (the luciferase reporter plasmid), and pTRANS230d (the T-DNA destination plasmid with the BASTA selection gene; Addgene #91113).

The Plasmid for luciferase expression in HEK293T cells was created by inserting the firefly luciferase gene into the lentiviral cloning vector pCDH-Neo via XbaI and EcoRI sites. The CRISPR-Cas9 plasmids (pTW224 and pTW225) targeting the luciferase gene were created by inserting CRISPR gRNA into pX458-GFP (pTW184) via BbsI digestion and ligating annealed oligos containing complementary overhangs (Supplementary Data 3)

### Plant transformation for CRISPR-Cas9 expression
All Arabidopsis transgenic plants were generated using the Columbia ecotype (Col-0). T-DNA-mediated floral dip transformation was performed according to the protocol as previously outlined[42]. Transgenic T1 seeds were sown on soil and treated with BASTA selection to select transgenic plants as previously described[42]. In brief, developing Arabidopsis inflorescences were simply dipped for a few seconds into a 5% sucrose solution containing 0.05% Silwet L-77 and resuspended Agrobacterium cells carrying the genes to be transferred. Whole plants were used for genotyping at 3 weeks post-germination using the amplicon-based next-generation sequencing assay.

*Arabidopsis* and *Setaria* protoplast transfection was performed by following the protocol described previously[19]. In brief, leaves from 14-day-old seedlings were sliced and digested in the enzyme solution (1.5% Cellulase, 0.75% Macerozyme, Kanematsu USA Inc.) for 4–5 h on a shaker at 40 rpm. Digested tissues were filtered through a 70uM nylon filter (Fisher Scientific LLC) into W5 buffer (2 mM MES with pH5.7, 154 mM NaCl, 125 mM CaCl$_2$, 5 mM KCl). Protoplasts were collected and resuspended in W5 buffer with a gentle centrifuge at 100$g$ for 5 min. Approximately 200,000 protoplasts were mixed with DNA plasmids (15 μg per construct) in 20% PEG buffer, incubated at room temperature in the dark for 48 h, then collected for the amplicon-based next-generation sequencing.

We adopted an Agrobacterium-mediated transient transformation approach, AGROBEST (Agrobacterium-mediated enhanced seedling transformation), for testing AtPolλ and its variants in the atpolλ-1 mutant. T-DNA constructs were assembled using the Golden Gate assembly method with pMOD_A0101 (the Cas9 expressing plasmid, Addgene #90998), the gRNA plasmids, the *AtUbi10* promoter driving *AtPolλ* coding sequences (either wild type or with amino acid variants), the AmCyan fluorescence reporter gene (Addgene #197731), and pTRANS230d (the T-DNA destination plasmid with the BASTA selection gene; Addgene #91113). The transient T-DNA transformation protocol was performed as described by Wu et al.[43] with the following modifications. The *Arabidopsis* seeds were germinated in 6-well plates in 1 mL of ½ MS liquid media with 0.5% sucrose (pH 5.5) in a growth incubator set to a 16-/8-h light/dark cycle, 22 °C, and 55% humidity. Thirteen-day-old seedlings were co-cultivated with the Agrobacterium strain EHA105 carrying each T-DNA construct at O.D. 0.02 for 3 days. The transformed plants were then transferred to 1 mL of ½ MS medium containing 250 ug/mL Timentin for 3 days under the same growth condition. Individual transformed plants underwent genomic DNA extraction followed by Next Generating Sequencing assay.

### Human cell culture, transfection, and lentivirus transduction

HEK293 and 293T cells (from ATCC with Cat# CRL-1573 and CRL-3216) were cultured in Dulbecco's modified Eagle's medium supplemented with 10% fetal bovine serum and 1% penicillin/streptomycin. For lentivirus production, 293T cells were co-transfected with pCDH carrying the luciferase gene and the packaging plasmids pCMV-ΔR8.91 and pMDG-VSV-G at a ratio of 3:1:3 using PEI STAR™ (Tocris Bioscience, Minneapolis, Minnesota, USA), a polyethylenimine (PEI) transfection reagent, following the manufacturer's instruction. Medium-containing virus particles were collected 72 h after transfection and applied to HEK293 cells for 48 h. Transduced HEK293 cells were then selected using G418 (800 μg/ml) for 5 days to establish a cell line stably expressing luciferase. HEK293-luciferase cells were transfected with pX458 carrying CRISPR gRNA using TransIT-X2 (Mirus Bio) based on the manufacturer's instruction. 48 h after the transfection, cells were harvested to extract genomic DNA for the amplicon-based next-generation sequencing.

### Mutation genotyping and characterization of mutation profiles

Total genomic DNA was extracted from the *Setaria* protoplast or *Arabidopsis* whole plant samples using the Cetyl trimethylammonium bromide (CTAB) method as described earlier[9,16,17]. Polymerase chain reaction (PCR) was performed from the genomic DNA with the oligonucleotides listed in Supplementary Data 3 by using GoTaq Green Mastermix (Promega Corp., Madison, Wisconsin, USA). The PCR conditions were followed as per the manufacturer's instruction with an annealing temperature at 55 °C (CHLI2 and MCsite_G) or 60 °C (MCsite_T) and an extension time of 1 min. Individual PCR amplicons were subjected to Illumina paired-end read sequencing (Genewiz Inc., South Plainfield, New Jersey, USA). The paired-end NGS reads were

analyzed using CRISPResso2 (version 2.1.1) to identify NHEJ-mediated indel mutations[44] with the default settings. The overall mutagenesis rates for each site were summarized in the Source Data file and Supplementary Data 4. The corresponding output files were loaded in R studio (Version 2023.09.1) for data plotting.

### Statistics and reproducibility

An unpaired one-tailed Student's $t$ test was conducted using R studio (Version 2023.09.1) to assess the significance of the mean value differences between the Polλ mutant or overexpression plants and the wild-type control groups. No statistical method was used to predetermine the sample size. No data were excluded from the analyses. For each experiment, all samples were randomly chosen. The treatments were compared to a control treatment without any prior knowledge of whether the experimental variables being altered would have a positive or negative impact on the results.

### Phylogenetic analyses

All X-family DNA polymerase protein sequences were obtained from NCBI GenBank (Supplementary Data 2). The neighbor-joining tree was generated using MEGA (version 11) with the default setting using the yeast DNA Pol4 as the outgroup[45].

### Reporting summary

Further information on research design is available in the Nature Portfolio Reporting Summary linked to this article.

## Data availability

The plasmids used in the Golden Gate assembly are available from Addgene with the ID number listed in the Methods. The Next-Generation sequencing data generated in this study have been deposited in the NCBI Sequence Read Archive (SRA) under accession code PRJNA1062985. Sequence data used from previously published article can be found in NCBI SRA under accession code PRJNA795172. The processed CRISPResso2 output files for each sample are available from github [https://github.com/ZhangLab-UMN/Weiss_Kumar_1bp_manuscript.git]. Other data generated in this study are provided in the Supplementary Information/Source Data file. Source data are provided with this paper.

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

## Acknowledgements

J.K. and F.Z. are partially supported by the National Science Foundation (IOS-2040218 and IOS-2206920) awards. T.W. was supported by a Bernard and Jean Phinney Graduate Fellowship in Plant Biology and a Doctoral Dissertation Fellowship from the University of Minnesota. C.C. and K.L. are supported by the Department of Defense CDMRP research award (W81XWH2010214). We thank all the Zhang lab members for their inputs.

## Author contributions

T.W., J.K., K.L., and F.Z. conceived and planned the study. T.W., J.K., C.C., S.G., J.L., O.S., and K.L. performed the experiments. F.Z., T.W., S.G., and J.K. analyzed the data. F.Z., T.W., J.K., and K.L. wrote the manuscript with input from all authors.

## Competing interests

The authors declare no competing interests.
