## [Peer Review File · Nature Communications]

Dual Activities of an X-family DNA Polymerase Regulate CRISPR-Induced Insertional Mutagenesis Across SpeciesREVIEWER COMMENTS

Reviewer #1 (Remarks to the Author):

I read the manuscript of White et al. with great interest. It is a very common phenomenon to find repair junctions with 1bp insertions after DSB induction by Cas9 in different eukaryotes. The general accepted explanation is that Cas9 cutting not only results in blunt ended DNA but also in a one nt 5' overhang, that might be filled in, resulting in 1 bp insertions. It was shown that X family polymerase are involved in this fill-in reaction in mammals.

The group of Feng Zhang analyzed DSB repair in plants namely in the dicot Arabidopsis as well as the monocot Setaria. Interestingly, the analysis demonstrated a major difference in the inserted nucleotides in case of iPAM_G sites in contrast to iPAM_T sites. In mammals mainly G insertions are found at iPAM_G sites, whereas in both plant species there are T or A insertions. Thus, plants have a higher incidence of non-template insertions.

The next question to answer was which enzymes are responsible for templated as well as non-templated insertions in plants. As it was known from mammals that Pol X-family polymerase are involved in the fill in reaction the authors knocked out Pol lambda, the only Pol X family member present in plant genomes. Surprisingly, in the mutant background templated but also non-templated insertions were massively reduced. On the other side, overexpression of Pol lambda enhanced templated and non-templated insertions. Thus, in plants Pol lambda is not only - like in mammals involved in the formation of templated insertions but is also able to transfer a single nucleotide onto a blunt ended DNA. Thus, Pol lambda of plants seems to have an efficient terminal desoxynucleotide transferase activity, also inline with the fact that no other TdT ORF is found in plant genomes. The authors were also able to identify a TdT specific AS signature in the plant Pol lambda ORF.

I see no faults in the experiments performed by the authors and totally agree with their conclusions. The manuscript is definitely interesting for a wide audience not only to learn that plants and mammals differ in the repair of Cas9 induced breaks but also which enzyme is responsible for this difference.

Reviewer #2 (Remarks to the Author):

In their manuscript entitled « Dual activities of an X-family DNA polymerase regulate CRISPR-induced insertional mutagenesis across species », Weiss and colleagues have investigated the outcome of CRISPR-Cas9 mutagenesis in two plant species, to determine the predictability of the induced mutations. They find that unlike what has been reported in animals, the predictability of the insertion pattern obtained in plants is low, and that this predictability depends on the nucleotide present next to the break site. Further, using mutant and over-expression lines, they show that insertions depend on the activity of Pol lambda, and that this enzyme appears to function both in a templated and in a non-templated manner. Overall, this work give interesting insight into plant CRISPR mutagenesis, and provides compelling experimental evidence supporting the conclusions regarding the patterns of mutations caused by CRISPR, and the role of Pol lambda in this process. The molecular determinants of these differences of behaviours between plant and Human Pol lambda are nicely discussed. Whether these results will allow better predicting the outcome of CRISPR mutagenesis is not as clear, because some important pieces of information are missing in the text (see below). In my opinion, additional experiments should be performed to strengthen the authors conclusions, both on the mechanistical point of view, and for the application of their findings, in order to warrant publication in a generalist journal such as Nature Communications.

- Throughout this study, authors chose to focus on insertions and not to consider deletions or substitutions. Although their choice is well justified, it would be important for the reader to know what proportion of induced mutations are indels, and how many of these are 1bp insertions.
- If 1bp insertions represent the majority of mutations, as authors seem to suggest, then this study could indeed pave the way for better prediction of CRISPR mutagenesis. If not, this point should not be so much emphasized in the article. According to Sup Figure S2, only about 30% of indels are 1bp

insertions at T-sites in *Setaria*. Considering that predicting the nature of the inserted nucleotide appears to be challenging, the predictability of such mutagenesis approach would remain very limited. Here authors do not take advantage of their results to propose a new prediction tool, which is a bit of a pity, as it would increase the interest of their findings for a broad audience. In addition, *Setaria* and *Arabidopsis* appear to behave differently at least in some instances, so such a tool would likely have to be optimized for each plant species.

- As mentioned above, the discussion about the dual activity of Pol lambda as TdT and Polymerase and the amino-acid changes that may explain this observation is very interesting. It would have been nice however to experimentally test this hypothesis.

- I am missing a discussion about the reasons why the nature of the nucleotide in the -4 position is so important for the nature of the inserted nucleotide. According to the model shown in Figure 5, it would seem that the choice between templated and non-templated activity would depend on whether the break generated staggered or blunt ends. But if this is so, why would that depend on the -4 nucleotide ?

- On Figure 3 X axis, what do numbers correspond to ?

- Authors may want to discuss the frequencies of inserted nucleotides when insertions are not templated. For example, on Figure 3B, insertion of A, T and G occur more or less at similar frequencies, but C is much less likely. C is also very rare on Figure S4. Why is that ?

Minor points

- DNA pol lambda is not only involved in the repair of UV-B induced lesions, it is also known to bypass oxidative damage (DOI: 10.1105/tpc.110.081455)

- In the methods section: "Whole plants were sacrificed for genotyping" is a bit of an odd choice of words. I would simply write used.

Reviewer #3 (Remarks to the Author):

I read this manuscript with great interest. The authors investigated the 1-bp insertion profiles in two plant species as well as human cells. They showed that 1-bp insertions in plants cannot be readily predicted by computational prediction tools that largely rely on template-based DNA repair. Based on the literature, the authors made a reasonable hypothesis that the DNA Pol λ gene is responsible for template-based NHEJ repair. With data in *Arabidopsis* mutant and overexpression lines, they provide strong genetic evidence for that. Further, they showed DNA Pol λ is responsible for both non-template and templated repair of DSBs in *Arabidopsis*, which is likely applicable to other plant species as well. This work made basic research discovery which has a potential translational impact on improving precise genome editing in plants and beyond. It is very straightforward and well-written. I have two comments here.

1) The authors should be careful in making conclusions by comparing the plant protoplast data with human cell culture data. For plant protoplasts, they are largely non-dividing, especially given the short window of the experiments done in the study. However, human cells are actively dividing. So, the authors were comparing non-dividing cells and dividing cells. As we know, the cell cycle could greatly shape DNA repair pathways. For example, the HR pathway is more active in replicating/dividing cells while the NHEJ pathways are more active in non-dividing cells. Cell cycles could also influence the expression of DNA Pol λ gene, regardless in plants or human cells. Probably, using cultured dividing plant cells would be a more appropriate means for the comparison here. I am not asking the authors to do such an experiment. However, it is important to realize these variables as they could provide alternative explanations for the observed data. The authors should reflect on their data and discuss this issue.

2) Along this line, the description of the transgenic *Arabidopsis* with CRISPR-Cas9 constructs in the WT and *atpolλ-1* mutant is not clear in the manuscript. How many transgenic T1 lines were analyzed for each construct? What is the genotype of each plant (homozygous, heterozygous, or mosaic)?

Providing such details/data points is necessary and may help partly address the question that I raised above on potential cell cycle-related DNA repair preference due to differential expression of DNA repair genes such as the DNA Pol λ gene.

Reviewer #1 (Remarks to the Author):

I read the manuscript of White et al. with great interest. It is a very common phenomenon to find repair junctions with 1bp insertions after DSB induction by Cas9 in different eukaryotes. The general accepted explanation is that Cas9 cutting not only results in blunt ended DNA but also in a one nt 5' overhang, that might be filled in, resulting in 1 bp insertions. It was shown that X family polymerase are involved in this fill-in reaction in mammals.

The group of Feng Zhang analyzed DSB repair in plants namely in the dicot Arabidopsis as well as the monocot Setaria. Interestingly, the analysis demonstrated a major difference in the inserted nucleotides in case of iPAM_G sites in contrast to iPAM_T sites. In mammals mainly G insertions are found at iPAM_G sites, whereas in both plant species there are T or A insertions. Thus, plants have a higher incidence of non-template insertions.

The next question to answer was which enzymes are responsible for templated as well as non-template insertions in plants. As it was known from mammals that Pol X-family polymerase are involved in the fill in reaction the authors knocked out Pol lambda, the only Pol X family member present in plant genomes. Surprisingly, in the mutant background templated but also non-templated insertions were massively reduced. On the other side, overexpression of Pol lambda enhanced templated and non-templated insertions. Thus, in plants Pol lambda is not only - like in mammals involved in the formation of templated insertions but is also able to transfer a single nucleotide onto a blunt ended DNA. Thus, Pol lambda of plants seems to have an efficient terminal deoxynucleotide transferase activity, also inline with the fact that no other TdT ORF is found in plant genomes. The authors were also able to identify a TdT specific AS signature in the plant Pol lambda ORF.

I see no faults in the experiments performed by the authors and totally agree with their conclusions. The manuscript is definitely interesting for a wide audience not only to learn that plants and mammals differ in the repair of Cas9 induced breaks but also which enzyme is responsible for this difference.

Response: We appreciate the reviewer's positive and encouraging comments on this research work.

Reviewer #2 (Remarks to the Author):

In their manuscript entitled « Dual activities of an X-family DNA polymerase regulate CRISPR-induced insertional mutagenesis across species », Weiss and colleagues have investigated the outcome of CRISPR-Cas9 mutagenesis in two plant species, to determine the predictability of the induced mutations. They find that unlike what has been reported in animals, the predictability of the insertion pattern obtained in plants is low, and that this predictability depends on the nucleotide present next to the break site. Further, using mutant and over-expression lines, they show that insertions depend on the activity of Pol lambda, and that this enzyme appears to function both in a templated and in a non-templated manner. Overall, this work give interesting insight into plant CRISPR mutagenesis, and provides compelling experimental evidence supporting the conclusions regarding the patterns of mutations caused by CRISPR, and the role of Pol lambda in this process. The molecular determinants of these differences of behaviours between plant and Human Pol lambda are nicely discussed. Whether these results will allow better predicting the outcome of CRISPR mutagenesis is not as clear, because some important pieces of information are missing in the text (see below). In my opinion, additional experiments should be performed to strengthen the authors conclusions, both on the mechanistical point of view, and for the application of their findings, in order to warrant publication in a generalist journal such as Nature Communications.

- Throughout this study, authors chose to focus on insertions and not to consider deletions or substitutions. Although their choice is well justified, it would be important for the reader to know what proportion of induced mutations are indels, and how many of these are 1bp insertions.

Response: Thank you for bringing up this point. To address this concern, we have included additional characterization of the indel frequencies and profiles for all 59 CRISPR sites tested in Figure 1 and 2A. The proportions of CRISPR induced mutation (indel mutagenesis rates) of individual CRISPR sites were summarized in Supplementary Figure S1a, averaging 8.9% for Arabidopsis and 28.4% for Setaria. Additionally, we conducted new analyses on the frequencies of individual insertion and deletion types for each target site (Figure 1a).

We have made the following changes in the manuscript:

1. New Supplementary figure S1a was included to summarize the CRISPR-Cas9 induced indel frequencies across 59 targeted sites in Arabidopsis and Setaria.

2. New main Figure 1a was added to show the frequencies of individual insertion and deletion types for each target site in Arabidopsis and Setaria.
3. New text was added in line 114-117 to summarize these results: “The indel mutagenesis rates averaged at 8.9% and 28.4% at the sites from Arabidopsis and Setaria, respectively (Supplementary Figure S1a). Additionally, the indel profile from each site was further characterized into individual insertion and deletion types for each species (Figure 1a).”
4. A new Supplementary Table 4 was included to summarize the indel mutagenesis rates for all the sites tested in this study.

• If 1bp insertions represent the majority of mutations, as authors seem to suggest, then this study could indeed pave the way for better prediction of CRISPR mutagenesis. If not, this point should not be so much emphasized in the article. According to Sup Figure S2, only about 30% of indels are 1bp insertions at T-sites in Setaria. Considering that predicting the nature of the inserted nucleotide appears to be challenging, the predictability of such mutagenesis approach would remain very limited. Here authors do not take advantage of their results to propose a new prediction tool, which is a bit of a pity, as it would increase the interest of their findings for a broad audience. In addition, Setaria and Arabidopsis appear to behave differently at least in some instances, so such a tool would likely have to be optimized for each plant species.

Response: We appreciate the reviewer for these insightful comments. One of the key findings of our study is the substantial variation of profiles of 1-bp insertion rates across different sequences and species. To emphasize this point, we conducted new analyses on the frequencies of individual insertion and deletion types across all 59 targeted sites in Arabidopsis and Setaria as summarized in Figure 1a. In Arabidopsis, 1-bp insertions represented the most prevalent mutation types, accounting for an average of 44.6% of all mutations across 26 CRISPR sites (Figure 1a). On the other hand, for Setaria viridis, the average 1-bp insertion rate seemed to be the 4th highest at 9.6% across 33 CRISPR sites (Figure 1a). Thus, these two plant species may exhibit distinct capacities for generating CRISPR-Cas9 induced 1-bp insertions. Since DNA Polymerase λ was identified as responsible for 1-bp insertions, the disparity in 1-bp insertion frequencies between Arabidopsis and Setaria suggests potential variations in the regulation, recruitment, or enzymatic properties of Pol λ during the repair process in these organisms.

Moreover, our data indicated that DNA Polymerase λ could possess distinct template-dependent properties for installing 1-bp insertions in different organisms. While the

Arabidopsis and Setaria homologs exhibited both template-dependent and independent activities, human DNA Polymerase λ appeared to primarily rely on template-dependent activity. These findings collectively underscore that a universally applicable prediction tool for all species is unlikely to be feasible. As the reviewer noted, such tools would need to be tailored and optimized to accurately predict both mutation rates and profiles for each individual species, accounting for the inherent differences in the enzymatic properties and activities of the DNA repair machinery.

Furthermore, our study proposed a novel strategy to improve the predictability of CRISPR-Cas9 induced mutations across species. By modulating the activity of DNA Pol λ , we could potentially address the inherent differences in DNA repair machinery among organisms. In Arabidopsis plants overexpressing Pol λ , we demonstrated that the frequency of 1-bp insertions could be substantially increased at CRISPR target sites. We are currently investigating whether directly recruiting Pol λ to the targeted genomic loci could further augment 1-bp insertion frequencies, thereby improving the predictability of mutagenesis outcomes. While systematic evaluation of recruitment strategies was beyond the scope of this research, this approach warrants further investigation, as it could pave the way for more predictable CRISPR-Cas9 mutagenesis across diverse species by circumventing the intrinsic variability in DNA repair machinery among different organisms.

To clarify these points, we made the following changes:

1. The data from the new main Figure 1a was summarized in line 117-125:
“Notably, the 1-bp insertions represent one of the most common occurring mutation types, as previously observed in human cell lines. In Arabidopsis, 1-bp insertions were the most prevalent mutation types, accounting for an average of 44.6% of all mutations across 26 CRISPR sites (Figure 1a). For Setaria viridis, the average 1-bp insertion rate appeared to be the 4th highest at 9.6% across 33 CRISPR sites (Figure 1a).”
2. The new sentences were added in the discussion (line 501-507) to emphasize the point that prediction tools would need to be tailored and optimized to accurately predict both mutation rates and profiles with high confidence for each individual species: “The varying rates and profiles of 1-bp insertions observed across species could stem from distinct properties of DNA Pol λ activity or its differential involvement in repairing CRISPR-Cas9 induced double-strand breaks. This underscores how inherent variability in DNA repair pathways and the enzymatic properties of key repair proteins, such as DNA Pol λ , contribute to the

species-specific CRISPR-Cas9 editing profiles. Consequently, enhancing the predictability of CRISPR-Cas9 mutagenesis prediction tools necessitates tailoring and optimizing these tools for individual species.”

3. The new sentences were added in the discussion (line 512-519) to emphasize the strategy to improve the predictability of CRISPR-Cas9 induced mutations across species by modulating the activity of DNA Pol λ : “In Arabidopsis plants overexpressing DNA Pol λ , we have demonstrated substantial increases in the frequency of 1-bp insertions at CRISPR target sites. Further investigation is warranted on whether directly recruiting DNA Pol λ to targeted genomic loci could further augment 1-bp insertion frequencies, thereby improving the predictability of mutagenesis outcomes. This strategy could pave the way for more predictable CRISPR-Cas9 mutagenesis across diverse species by circumventing the intrinsic variability in DNA repair pathway machinery among organisms.”

- As mentioned above, the discussion about the dual activity of Pol lambda as TdT and Polymerase and the amino-acid changes that may explain this observation is very interesting. It would have been nice however to experimentally test this hypothesis.

Response: To elucidate the dual activities of plant DNA Polymerase Pol λ , we conducted additional experiments testing amino acid substitutions within the conserved SY/GF and YF/AW motifs (Supplementary Figure S6a). We generated two AtPol λ variants: AtPol λ _YF (A459Y/W460F), resembling human Pol λ by substituting amino acids AW with YF, and AtPol λ _GF (S366G/Y367F), resembling human TdT by substituting amino acids SY with GF (Figure 6a). These variants were expressed alongside the CRISPR-Cas9 construct in Arabidopsis DNA Pol λ knock-out plants using the stable transgenic approach.

Our new data revealed the significant changes in the insertion profiles, but not deletions, with each individual variant (Figure 6b and c, Supplementary Figure S6b, c and d). Specifically, compared to the overexpression of the wildtype AtPol λ , the AtPol λ _YF variant significantly increased the proportion of templated insertions by 100%, while the AtPol λ _GF variant significantly increased the proportion of non-templated insertions by 18%, aligning with our dual activity model. Notably, the overall insertion rates with each variant decreased to 4.4% and 5.5%, an 83-84% reduction compared to the overexpression of the wildtype AtPol λ (Supplementary Figure S6e). This suggests involvement of additional amino acids in regulating enzymatic activity. Further investigation is warranted to improve the enzymatic activities of these variants.

To clarify these points, we made the following changes:

1. New Figure 6a-c and Supplementary Figure S6a-e were added to summarize the AtPol λ variant testing experiments.
2. A whole new result section was added to describe these experiments in line 305-355 as follows:

“Conserved motifs modulate the dual activities of AtPol λ ”

To gain insights into the mechanism(s) underpinning the distinct properties of DNA Pol λ across species, we conducted protein sequence analyses by aligning AtPol λ with X-family DNA Polymerases in humans (Supplementary Figure S5). Previous studies have indicated two conserved motifs in human X-family DNA Polymerase that contribute to template dependency²³. The first motif, identified as GSYRRG in template-dependent human DNA polymerases λ , features two amino acids, serine and tyrosine (SY), which are replaced by glycine and phenylalanine (GF) in the template-independent human TdT (Figure 6a and Supplementary Figure S5)^{23,29}. The second motif, known as the YF motif, contains tyrosine and phenylalanine at the catalytically active sites of the DNA polymerases λ . In contrast, these two residues are changed to glycine and tryptophan (GW) in TdT (Supplementary Figure S5)^{23,29}. When analyzing these motifs in DNA polymerase λ homologs from Arabidopsis, Setaria, Tobacco, and rice, the first motif was identical to the sequences in human DNA Pol λ , while the second motif, characterized by alanine and tryptophan (AW), showed a closer resemblance to the GW motif found in human TdT (Figure 6a and Supplementary Figure S5). Thus, the plant DNA Pol λ homologs appear to combine characteristic motifs from human DNA Pol λ and TdT.

The presence of both human DNA Pol λ and TdT motifs could potentially contribute to the observed dual templated-dependent and independent activities in AtPol λ . We then hypothesized that the dual activities of AtPol λ could be modulated by modifying each motif individually. To test this hypothesis, we generated two variants of AtPol λ through site-directed mutagenesis on the respective motifs. The first variant, AtPol λ _YF (A459Y/W460F), was engineered by substituting Alanine and Tryptophan (AW) with Tyrosine and Phenylalanine (YF) at the second motif to mimic human DNA Pol λ (Figure 6a). Similarly, the second variant, AtPol λ _GF (S366G/Y367F), was created to mimic human TdT by replacing Serine and Tyrosine (SY) with Glycine and Phenylalanine (YF) at the first motif (Figure 6a).

The coding sequence of each At DNA Pol λ variant was cloned into the T-DNA vector described above, with the constitutive Arabidopsis Ubiquitin-10 promoter and a CRISPR-Cas9 expression cassette to target the CHLI2 site. We used an agrobacterium-mediated transient expression approach to transform individual T-DNA constructs into young seedlings of the *atpol λ* knock-out mutant, and then examined the CRISPR-Cas9 mutation profile at the CHLI2 site using the NGS assay (Supplementary Figure S6a). The average mutation rates from these samples are 17.3% (the AtPol λ wild type, i.e. AtPol λ _WT), 12.7% (AtPol λ _GF) and 10.3% (AtPol λ _YF), respectively (Supplementary S6b). When analyzing templated versus non-templated 1-bp insertion patterns, the samples expressing the wild type AtPol λ

gene exhibited higher proportions of non-templated insertions compared to those of templated insertions (57.6 % non-templated insertions versus 42.4% templated insertions) consistent with the observations from the stable transgenic plants (Figure 5a and 6b). In contrast, the samples transformed with the AtPol λ _YF variant demonstrated altered 1-bp insertion profiles with templated insertion proportions being significantly higher than those from the overexpression of the wildtype AtPol λ by 100% (86.0% versus 42.4%; Figure 6b and Supplementary Figure S6c and d). Conversely, the samples transformed with the AtPol λ _GF variant displayed significantly higher proportions of non-templated insertions compared to the wild-type AtPol λ overexpression lines by 18% (67.9 % versus 57.6%; Figure 6b and Supplementary Figure S6c and d). Regarding the deletion profiles, no evident differences were observed within three different deletion groups, 1-bp, 2 to 10-bp and more than 10-bp, among AtPol λ _WT and the two variants (Figure 6c).

Notably, the overall 1-bp insertion rates from the samples with each variant reduced to 4.4% and 5.5% compared to 31.7% in the wild-type AtPol λ overexpression control, suggesting involvement of additional amino acids in regulating enzymatic activity. (Supplementary Figure S6e). Collectively, these observations align with our hypothesis that these two conserved motifs play crucial roles in modulating the dual template-dependent and independent activities of AtPol λ . Further investigation is required to refine the enzymatic activities of these variants.”

3. We modified the text in the discussion to incorporate insights from our new experiments.

The original text:

“Protein sequence analysis could provide additional insights into the distinct properties of DNA Pol λ across species. When we examined the catalytic sites in X-family DNA polymerases, two conserved motifs have been suggested to contribute to the template dependency²³. The first motif, identified as GSYRRG in template-dependent DNA polymerases λ , features serine and tyrosine (SY) replaced by glycine and phenylalanine (GF) in TdT (Supplementary Figure S5)^{23,29}. The second motif, known as the YF motif, features tyrosine and phenylalanine at the catalytically active sites of template-dependent DNA polymerases λ . In contrast, these two residues are replaced by glycine and tryptophan (GW) in the non-template dependent X-family DNA polymerases, TdT (Supplementary Figure S5)^{23,29}. When analyzing these motifs in DNA polymerase λ homologs from Arabidopsis, Setaria, Tobacco, and rice, the first motif was shown to be identical to the sequence in human DNA Pol λ , while the second motif, characterized by alanine and tryptophan (AW), showed a closer resemblance to the GW motif found in human TdT (Supplementary Figure S5). The plant DNA Pol λ homologs seem to possess characteristic motifs from both human DNA Pol λ and TdT, which could explain its dual activities in generating both templated and non-templated insertions.”

The modified text (line 396-404):

“Furthermore, protein sequence analysis revealed that plant DNA Pol λ homologs possess characteristic motifs from both human DNA Pol λ and TdT, which could elucidate the observed dual activities of AtPol λ . Our in vivo assay with the AtPol λ variants, generated by modifying these motifs, further supports this hypothesis. The AtPol λ variant resembling human DNA Pol λ significantly increased the proportion of templated insertions, while the human TdT-like variant significantly increased the proportion of non-templated insertions. These findings suggest that the dual activities of plant DNA Pol λ stem from the combination of conserved motifs from both template-dependent and independent polymerases.”

- I am missing a discussion about the reasons why the nature of the nucleotide in the -4 position is so important for the nature of the inserted nucleotide. According to the model shown in Figure 5, it would seem that the choice between templated and non-templated activity would depend on whether the break generated staggered or blunt ends. But if this is so, why would that depend on the -4 nucleotide ?

Response: Thank you for raising this question. According to our model, the choice between templated and non-templated activities depends on the generation of staggered or blunt ends in plants. However, the ratio between staggered and blunt cleavage can vary in different CRISPR sites. Recent studies in human cell lines have suggested that the sequence context, particularly the nucleotide at the -4 position, could play a crucial role in influencing the ratio of blunt and staggered cleavages. For example, target sites with a thymine (T) at the -4th position appeared to exhibit higher staggered cleavage compared to those with other nucleotides (DOI: 10.1038/nbt.4317, DOI: 10.1038/s41467-019-09551-w). In our model, a higher frequency of staggered cleavages would result in more templated insertions, and vice versa. It is important to note that while the nucleotide at the -4 position has been identified as a key contributing factor, it is likely that additional sequence features or other factors could influence the ratio of blunt and staggered cleavages. To systematically explore this question further, experiments employing approaches to capture double-strand break (DSB) end structures in vivo could be employed (DOI: 10.1101/2023.01.10.523377).

To clarify these points, we modified the following sentences in the discussion (line 431-477).

The original text:

“Additionally, this model suggests that the ratios between blunt versus staggered cleavage products determine the templated versus non-templated insertion ratios at individual target sites. In line with this, recent studies have suggested that the sequence context, particularly the -4th nucleotide, could play a crucial role in influencing the blunt and staggered cleavage dynamics^{11,32}.”

The modified text:

“Moreover, the ratio between staggered and blunt cleavage can vary in different CRISPR sites. Recent studies in human cell lines have suggested that the sequence context, particularly the nucleotide at the -4 position, could play a crucial role in influencing the ratio of blunt and staggered cleavages. For example, target sites with a thymine (T) at the -4 position appeared to exhibit higher staggered cleavage compared to those with other nucleotides^{11,32}. In our model, a higher frequency of staggered cleavages would result in more templated insertions, and vice versa. While the nucleotide at the -4 position has been identified as a key contributing factor, it is likely that additional sequence features or other factors could influence the ratio of blunt and staggered cleavages. To systematically explore this question further, experiments employing approaches to capture double-strand break (DSB) end structures in vivo could be employed³³.”

- On Figure 3 X axis, what do numbers correspond to ?

Response: Thank you for bringing this to our attention. The X-axis indicates the individual sites of MCSite_T and MCSite_G. We added the label in Figure 3 and the clarifications in the figure legend accordingly.

- Authors may want to discuss the frequencies of inserted nucleotides when insertions are not templated. For example, on Figure 3B, insertion of A, T and G occur more or less at similar frequencies, but C is much less likely. C is also very rare on Figure S4. Why is that ?

Response: The lower frequency of cytosine (C) insertions compared to adenine (A), thymine (T), and guanine (G) insertions in the non-templated cases (Figure 3B and Supplementary Figure S4) is indeed noteworthy. One possible explanation could be that the Arabidopsis DNA Pol λ has preferences for certain types of nucleotides when installing non-templated insertions. Substrate nucleotide preferences have been previously reported in human Terminal deoxynucleotidyl transferase (TdT), which catalyzes non-templated nucleotide additions. In vitro studies have shown that human TdT exhibits a preference for incorporating nucleotides in the order of G > T = A > C (DOI: 10.3390/ijms25020879, DOI: 10.26508/lsa.202201428). It is plausible that Arabidopsis Pol λ could also exhibit nucleotide preferences, potentially favoring G, T, and A over C during non-templated insertions in vivo. To explore this hypothesis, further experiments would be required.

To clarify this point, we have included the following sentences in line 405-421:

“Moreover, it is worth noting that the frequencies of a specific inserted nucleotide can vary for non-templated insertions. For example, lower frequencies of cytosine (C) insertions compared to adenine (A), thymine (T), and guanine (G) were observed in the CHLI2 and MCSite_T sites (Figure 3B and Supplementary Figure S4). One possible explanation could be that the AtPol λ exhibits preferences for certain types of nucleotides when installing non-templated insertions. Substrate nucleotide preferences have been previously reported in human Terminal deoxynucleotidyl transferase (TdT), which catalyzes non-templated nucleotide additions. In vitro studies have shown that human TdT exhibits a preference for incorporating nucleotides in the order of G > T = A > C^{31,32}. It is plausible that AtPol λ also displays nucleotide preferences, potentially favoring G, T, and A over C during non-templated insertions in vivo. To explore this hypothesis and elucidate the AtPol λ activity during non-templated synthesis, further experiments would be required.”

Minor points

- DNA pol lambda is not only involved in the repair of UV-B induced lesions, it is also known to bypass oxidative damage (DOI: 10.1105/tpc.110.081455)

Response: We have added the following text “repairing oxidative base damage” in line 376 and cited the literature accordingly.

- In the methods section: “Whole plants were sacrificed for genotyping” is a bit of an odd choice of words. I would simply write used.

Response: We have changed the wording to “used” accordingly in line 565.

Reviewer #3 (Remarks to the Author):

I read this manuscript with great interest. The authors investigated the 1-bp insertion profiles in two plant species as well as human cells. They showed that 1-bp insertions in plants cannot be readily predicted by computational prediction tools that largely rely on template-based DNA repair. Based on the literature, the authors made a reasonable hypothesis that the DNA Pol λ gene is responsible for template-based NHEJ repair. With data in Arabidopsis mutant and overexpression lines, they provide strong genetic evidence for that. Further, they showed DNA Pol λ is responsible for both non-template and templated repair of DSBs in Arabidopsis, which is likely applicable to other plant species as well. This work made basic research discovery which has a potential translational impact on improving precise genome editing in plants and beyond. It is very straightforward and well-written. I have two comments here.

1) The authors should be careful in making conclusions by comparing the plant protoplast data with human cell culture data. For plant protoplasts, they are largely non-dividing, especially given the short window of the experiments done in the study. However, human cells are actively dividing. So, the authors were comparing non-dividing cells and dividing cells. As we know, the cell cycle could greatly shape DNA repair pathways. For example, the HR pathway is more active in replicating/dividing cells while the NHEJ pathways are more active in non-dividing cells. Cell cycles could also influence the expression of DNA Pol λ gene, regardless in plants or human cells. Probably, using cultured dividing plant cells would be a more appropriate means for the comparison here. I am not asking the authors to do such an experiment. However, it is important to realize these variables as they could provide alternative explanations for the observed data. The authors should reflect on their data and discuss this issue.

Response: Thank you for raising this excellent point. We acknowledge the importance of being cautious when comparing CRISPR-Cas9 mutagenesis data between plant protoplasts and human cells, as they may undergo different cell cycle states. In this study, we should have clarified that the Arabidopsis mutagenesis data for this comparison were collected from stable transgenic plants, while the Setaria mutagenesis data were derived from protoplast transfection. In other words, the mutation profiles observed in Arabidopsis were derived from dividing cells, while only the data from Setaria were obtained from non-dividing protoplast cells.

As comparing CRISPR-induced mutagenesis across different cell states could introduce variability, it is indeed interesting to address the potential differences in 1-bp insertion profiles between dividing and non-dividing cells. One approach, as suggested by the reviewer, is to utilize cultured dividing Setaria cells to compare with the current data.

However, to our knowledge, no dividing *Setaria* cell lines were available in the research community. Alternatively, we could compare 1-bp insertion profiles between leaf samples from stable *Arabidopsis* transgenics (representing dividing cells) and protoplast samples (representing non-dividing cells). To achieve this, we conducted new experiments using *Arabidopsis* protoplasts to target three CRISPR sites: CHLI2, MCsite_T, and MCSite_G. We then compared the 1-bp insertion rates and profiles of these sites between the stable transgenic lines and protoplasts. As a result summarized in Supplementary Figure S6f and Table 4, we observed similar 1-bp mutation outcomes between the two groups. While our data suggested that the cell cycle state may not be a major determinant of the 1-bp insertion patterns observed in this study, it has been reported that environmental factors can regulate the expression level and protein stability of AtDNA Pol λ (DOI: 10.1371/journal.pone.0133843; DOI: 10.4161/psb.22715). Further investigation would be required to examine the CRISPR mutagenesis profiles under different growth conditions.

To clarify these points, we have made the following changes in the manuscript:

1. The following sentences were added in the Results to clarify whether the stable transgenic plants or protoplast cells were used in the CRISPR mutation assay. In line 108 -113: "In *Arabidopsis*, each CRISPR-Cas9 construct was introduced using the floral dip-based stable transgenic approach. Individual seedlings of each T1 transgenic plant were collected for the CRISPR mutation analysis at each target site. In *Setaria*, individual CRISPR-Cas9 constructs were transformed via transient protoplasts transfection. Transformed protoplast cells were collected after 48 hours for the mutation assay."
2. We added the new data in Supplementary Figure S6f and Supplementary Table 4 to illustrate the similar 1-bp insertion patterns between leaf samples from stable transgenics (representing dividing cells) and protoplast samples (representing non-dividing cells) in *Arabidopsis*.
3. We included the following sentences in the Discussion to reflect on the new data and how different cell cycle states could influence DNA repair outcomes. In line 483-496: "It is noteworthy that, in this study, we compared 1-bp insertions across different cell states from three different species, including non-dividing *Setaria* protoplast cells, dividing cells from *Arabidopsis* seedlings, and human dividing cell lines. It has been known that cell states, especially cell cycle stages, could influence DNA repair pathway choices and expression of DNA repair genes³⁶. To address the potential differences in 1-bp insertion profiles between dividing and non-dividing cells, we conducted additional experiments using *Arabidopsis* protoplasts to target three CRISPR sites: CHLI2, MCsite_T, and MCSite_G. When comparing the 1-bp insertion rates and profiles of these sites between the stable transgenic lines and protoplasts, we observed similar 1-bp mutation outcomes between the two groups (Supplementary Figure S6f and Supplementary Table 4). While our data indicate that the cell cycle state may not

be a major determinant of the 1-bp insertion patterns observed in this study, it has been reported that environmental factors can regulate the expression level and protein stability of AtPol λ ^{38,39}. Further investigation would be required to examine the CRISPR mutagenesis profiles under different growth conditions.”

2) Along this line, the description of the transgenic Arabidopsis with CRISPR-Cas9 constructs in the WT and atpol λ -1 mutant is not clear in the manuscript. How many transgenic T1 lines were analyzed for each construct? What is the genotype of each plant (homozygous, heterozygous, or mosaic)? Providing such details/data points is necessary and may help partly address the question that I raised above on potential cell cycle-related DNA repair preference due to differential expression of DNA repair genes such as the DNA Pol λ gene.

Response: We have implemented the following changes throughout the manuscript to describe transgenic Arabidopsis with the CRISPR-Cas9 construct in both wild type and atpol λ -1 mutant.

1. We included the description of stable transgenic Arabidopsis plants in the section of “Plant-specific 1-bp insertion profiles dependent on the -4th nucleotide context”.

Line 179-186: “In Arabidopsis, CRISPR-Cas9 constructs were assembled with the firefly luciferase reporter gene in T-DNA. The resulting constructs were transformed using the floral dip-based stable transgenic approach. Three seedlings from each T1 transgenic group were collected for CRISPR mutation analysis at each target site. For *Setaria viridis*, a homozygous *Setaria* line with the firefly luciferase reporter gene integrated into the genome was obtained from previous research¹⁹. Individual CRISPR-Cas9 constructs were then transformed into protoplast cells isolated from the luciferase gene-containing plants. Transformed protoplasts were collected after 48 hours for the mutation assay with 3 replications for each target site.”

2. The sentences were modified to include the information of transgenic Arabidopsis plants in the section “DNA Pol λ homolog responsible for both templated and non-templated 1-bp insertions in plants” (line 233-238).

Original text:

“Using this mutant line, we generated transgenic CRISPR-Cas9 plants to target three distinct sites: the single-copy site in the Arabidopsis Cheletase I2 gene (CHLI2), as well as the MCSite_T and MCSite_G sites.”

Modified text:

“Using the wild type and the homozygous *atpolλ-1* mutant Arabidopsis plants, we generated stable transgenic plants with the CRISPR-Cas9 T-DNA construct to target three distinct sites: the single-copy site in the Arabidopsis Cheletase I2 gene (*CHLI2*), as well as the MCSite_T and MCSite_G sites. Three T1 CRISPR-Cas9 transgenic plants from each genotype were used to survey CRISPR-induced mutations for each target site.”

3. The following sentences were added to include the information of transgenic Arabidopsis plants in the section “Overexpression of *AtPolλ* restores or enhances templated and non-templated 1-bp insertions”

Line 276-277:

“Three T1 CRISPR-Cas9 transgenic plants with the *atpolλ-1* mutant genotype were used to survey CRISPR-induced mutations for each target site.

Line 293-294:

“Three T1 CRISPR-Cas9 transgenic plants with the wild-type background were used to survey CRISPR-induced mutations for each target site.”

REVIEWERS' COMMENTS

Reviewer #2 (Remarks to the Author):

The authors have very nicely and thoroughly addressed all my comments regarding the original data. In addition, they have performed extra-experiments to functionally analyze the role of conserved motifs in the sequence of pol lambda, thereby strengthening the mechanistical aspect of the paper. This is a very nice piece of work, I have no further comments.

Reviewer #3 (Remarks to the Author):

The authors have adequately addressed my points. I have no more concerns and hence fully endorse its publication at Nature Communications.